# Metabolic Reprogramming of Glioblastoma Cells during HCMV Infection Induces Secretome-Mediated Paracrine Effects in the Microenvironment

**DOI:** 10.3390/v14010103

**Published:** 2022-01-07

**Authors:** Mark A. A. Harrison, Emily M. Hochreiner, Brooke P. Benjamin, Sean E. Lawler, Kevin J. Zwezdaryk

**Affiliations:** 1Neuroscience Program, Tulane Brain Institute, Tulane University School of Science & Engineering, New Orleans, LA 70118, USA; mharri26@tulane.edu (M.A.A.H.); ehochreiner@tulane.edu (E.M.H.); 2School of Liberal Arts, Tulane University, New Orleans, LA 70118, USA; bbenjamin@tulane.edu; 3Legoretta Cancer Center, Department of Pathology and Laboratory Medicine, Warren Alpert Medical School, Brown University, Providence, RI 02912, USA; sean_lawler@brown.edu; 4Department of Microbiology & Immunology, Tulane University, New Orleans, LA 70112, USA; 5Tulane Center for Aging, Tulane University, New Orleans, LA 70112, USA

**Keywords:** human cytomegalovirus (HCMV), glioblastoma (GBM), metabolism, aerobic glycolysis, oxidative phosphorylation (OXPHOS), lactate, reactive oxygen species, tumor microenvironment

## Abstract

Glioblastoma (GBM) is an aggressive primary central nervous system neoplasia with limited therapeutic options and poor prognosis. Following reports of cytomegalovirus (HCMV) in GBM tumors, the anti-viral drug Valganciclovir was administered and found to significantly increase the longevity of GBM patients. While these findings suggest a role for HCMV in GBM, the relationship between them is not clear and remains controversial. Treatment with anti-viral drugs may prove clinically useful; however, their results do not explain the underlying mechanism between HCMV infection and GBM progression. We hypothesized that HCMV infection would metabolically reprogram GBM cells and that these changes would allow for increased tumor progression. We infected LN-18 GBM cells and employed a Seahorse Bioanalyzer to characterize cellular metabolism. Increased mitochondrial respiration and glycolytic rates were observed following infection. These changes were accompanied by elevated production of reactive oxygen species and lactate. Due to lactate’s numerous tumor-promoting effects, we examined the impact of paracrine signaling of HCMV-infected GBM cells on uninfected stromal cells. Our results indicated that, independent of viral transmission, the secretome of HCMV-infected GBM cells was able to alter the expression of key metabolic proteins and epigenetic markers. This suggests a mechanism of action where reprogramming of GBM cells alters the surrounding tumor microenvironment to be permissive to tumor progression in a manner akin to the Reverse-Warburg Effect. Overall, this suggests a potential oncomodulatory role for HCMV in the context of GBM.

## 1. Introduction

Glioblastoma (GBM) is the most common primary malignant central nervous system neoplasia. Patients have a poor prognosis, with 5-year survival rates around 5% [1]. The current standard of care is tumor resection, irradiation, and treatment with the chemotherapy drug temozolomide [2]. Notably, patient survival has failed to improve significantly over the past decade emphasizing the urgent need for improved or novel therapies [1]. One field currently under investigation is the role of viruses in oncogenesis or oncomodulation. Several viruses have been closely associated with cancers and are identified as oncogenic accounting for approximately 15% of human cancer [3,4,5]. In the context of GBM, numerous neurotropic viruses have been investigated mainly in the polyomavirus and herpesvirus families due to their prevalence and persistence, but no direct association has been identified [6,7].

The presence of human herpesvirus-5 (cytomegalovirus (HCMV)) in GBM tumors and surrounding tissue was first reported by Cobbs and Britt in 2002 [8]. Since then, numerous groups have reported HCMV proteins and/or transcripts in GBM specimens [9,10,11,12,13,14,15,16]. However, there is much debate within the field as to the validity of these findings as other groups have found no evidence of HCMV in GBM tumors [17,18]. Despite the absence of a clearly defined role for HCMV in GBM progression, Phase 2 clinical trials were initiated adding valganciclovir (an anti-viral agent used to treat HCMV infection) to the current standard of care. Inclusion of valganciclovir increased median overall survival (24.1 vs. 13.3 months) and 2-year survival rate (49.8% vs. 17.3%) [19,20]. These findings support the hypothesis that HCMV plays a role in GBM but fail to elucidate a mechanism of action. The role of HCMV in cancer remains a polarizing topic. A current hypothesis suggests that HCMV may behave in an oncomodulatory capacity, altering the tumor microenvironment to promote tumor survival, expansion, and metastasis. These properties have been linked to multiple mechanisms including increased cell division, cell migration and angiogenesis. However, the effects of HCMV on tumor metabolism have not been examined. We hypothesize that metabolic reprogramming of host cells following HCMV infection may play a significant role in tumor progression.

HCMV is a ubiquitous herpesvirus with seroprevalence rates upwards of 60% in many populations [21,22,23]. Most primary infections are asymptomatic or result in flu-like symptoms followed by lifelong episodes of latency and reactivation. HCMV infection has been demonstrated to disrupt anti-viral signaling pathways [24], induce chromosomal aberrations by inhibiting DNA repair mechanisms [25], and inhibit the function of various immune cell populations [26] resulting in immune evasion. HCMV has also been reported to manipulate the metabolic functions of host cells, affecting, fatty acid oxidation [27], glycolysis [28,29], and oxidative phosphorylation (OXPHOS) [30,31,32]. Intriguingly, these consequences of HCMV infection, reprogramming of energy metabolism and evading immune destruction, represent the two “Emerging Hallmarks” of cancer as described by Hanahan and Weinberg [33]. Many cancer cells, despite the presence of adequate oxygen, upregulate aerobic glycolysis, commonly referred to as the Warburg Effect [34,35]. Interestingly, upregulation of aerobic glycolysis has been observed in fibroblasts following HCMV infection [28,29]. More recently, another related hypothesis posits that cancerous cells alter the surrounding stromal cells’ metabolism resulting in the Warburg Effect [36]. These cells produce metabolic products which are then transported to the cancer cells to accommodate their proliferation and survival in a process termed the Reverse Warburg Effect.

In this manuscript, we characterize the metabolic reprogramming that occurs following acute infection of GBM with HCMV in vitro. We provide evidence that infection produces a significant increase in the rate of host cell glycolysis and OXPHOS. Additionally, we demonstrate that these alterations lead to the production of metabolites which may significantly alter the tumor microenvironment. Finally, we demonstrate the ability of HCMV infected GBM cells to alter the levels of key metabolic proteins and induce epigenetic changes in uninfected neighboring cells. These data lay the foundation of a mechanism of HCMV-mediated tumor progression in the context of both GBM and cancers in general.

## 2. Methods

### 2.1. Cell and Virus Culture

Glioblastoma (LN-18) and human foreskin fibroblast (HFF) cell lines were purchased from American Type Culture Collection (ATCC #CRL-2610, #SCRC-1041; Manassas, VA, USA) and cultured in Dulbecco’s Modified Eagle Medium (DMEM; #11965-084; Thermo Fisher Scientific, Waltham, MA, USA) containing 4.5 g/L glucose and 4 mM L-glutamine supplemented with 5% or 10% fetal bovine serum (FBS; #S11550; Atlanta Biologicals, Flowery Branch, GA, USA) respectively. Cells were maintained at 37 °C with 5% CO_2_ and routinely tested for mycoplasma using the MycoAlert PLUS kit (#LT07-701; Lonza, Morristown, NJ, USA).

Viral stocks of HCMV Towne-GFP were propagated in HFF cells using a multiplicity of infection (MOI) of 0.1 as previously described [37]. Briefly, cells were cultured with the virus until ~90% cytopathic effect was observed. Remaining adherent cells were scraped from plates. Cellular debris and supernatant were pelleted by centrifugation. Clarified supernatant was added to thin-wall, ultra-clear 25 × 89 mm tubes (#344058; Beckman Coulter, Brea, CA, USA), a 20% sucrose solution was underlaid, and cells were centrifuged for 40 min at 104,000× *g* at 16 °C. The viral pellet was resuspended in tris sodium chloride (TN) buffer.

Viral titers were determined by serial titration of viral stocks on HFF cells in triplicate. Following 48 h of culture, media was removed, cells were washed with 1× PBS, and fixed with 95% ethanol. Cells were incubated with anti-HCMV IE1 mouse polyclonal antibody (1:200; Generously gifted by Bill Britt) for one hour at 37 °C followed by Alexa Fluor 594 goat anti-mouse polyclonal secondary antibody (1:1000; #A11005; Invitrogen Corp., Carlsbad, CA, USA) for 1 h at room temperature. Stained cells were counted manually using a Nikon Eclipse Ti2 microscope (Nikon Inc., Tokyo, Japan).

Viral infection was performed on confluent LN-18 cells using an MOI of 3 for 90 min. Mock-infected and infected cells were then washed with 1× PBS, fresh complete culture media was added, and cells were maintained for the indicated infection period.

### 2.2. xCELLigence Real-Time Cell Analysis

An Agilent xCELLigence platform (Agilent Technologies Inc., Santa Clara, CA, USA) was used to examine cytopathic effect following infection with HCMV. Cells were seeded into a 96-well E-plate at 80% confluence and Cell index (impedance) was measured at 15-min intervals. When Cell Index stabilized (~48 h post seeding) cells were infected with HCMV at an MOI of 3. Cell index was measured every 15 min for the duration of the experiment. Reads were normalized to the timepoint immediately prior to infection.

### 2.3. RNA Isolation and qRT-PCR

RNA was isolated using the RNeasy Minikit (#74106; Qiagen, Hilden, DE) and converted to cDNA using the iScript cDNA synthesis kit (#1708891; Bio-Rad, Hercules, CA, USA) according to the manufacturer’s instructions. qRT-PCR was conducted on cDNA using Sso Advanced SYBR Green Supermix (#1725271; Bio-Rad) and gene-specific primers.

Primers: SCL16A1 (MCT1; qHsaCID0008777; Bio-Rad); SLC16A3 (MCT4; qHsaCID0014322; Bio-Rad); LDHA (qHsaCED0001212; Bio-Rad) LDHB (qHsaCID0012068; Bio-Rad); HCMV IE1 (F: 5′-CAAGTGACCGAGGATTGCAA-3′, R: 5′-CACCATGTCCACTCGAACCTT-3′; IDT DNA Technologies, Coralville, IA, USA); RPL13A (F: 5′-CTCAAGGTCGTGCGTCTGAA-3′, R: 5′-TGGCTGTCACTGCCTGGTACT-3′; IDT DNA).

Cycle values were normalized to ribosomal protein L13a (RPL13A), and relative gene expression was determined using the 2^−∆∆Ct^ method and reported as fold change relative to untreated controls.

### 2.4. RNA-Seq

RNA isolation and sequencing were performed as previously described [38]. Briefly, G44 glioblastoma stem cells were mock- or HCMV-infected in triplicate and maintained for 72 HPI. RNA was isolated, assessed for degradation and contamination, and sequencing libraries were prepared using the Illumina paired-indexing protocol. STAR software was used to map raw reads to the reference organism. Data were cleaned and deposited in EMBL-EBI’s ArrayExpress database (E-MTAB-7613).

### 2.5. Seahorse Bioanalyzer

Metabolic profiling was conducted with XFe24 Seahorse Bioanalyzer (Agilent) according to the manufacturer’s instructions. Briefly, XFe24 cell plates were coated with Cell-Tak cell and tissue adhesive (#354240; Corning Inc., Corning, NY, USA) at a concentration of 3.5 μg/cm^2^. Cells were seeded at a concentration of 1.2 × 10^5^ cells per well and centrifuged at 450 rpm to permit adherence to the tissue adhesive. Oxygen consumption rate (OCR) and extracellular acidification rate (ECAR) were measured for at least three independent experiments and used to characterize the metabolic phenotype of the cells. *XF Real-Time ATP Rate Test* (#103592-100; Agilent) was used to determine total ATP production and ATP production derived from either glycolysis or oxidative phosphorylation (OXPHOS) following treatments with oligomycin (1.5 μM) and rotenone/antimycin A (0.5 μM). *XF Mito Stress Test* (#103015-100, Agilent) was used to examine mitochondrial metabolic function by injecting oligomycin (1 μM), FCCP (1 μM), and rotenone (0.5 μM) according to the manufacturer’s protocol. *XF Glycolysis Stress Test* (#103020-100, Agilent) was used to measure glycolysis by injecting glucose (10 mM), oligomycin (1 μM), and 2-deoxyglucose (100 mM) according to the protocol. At the completion of the Seahorse assay cells were stained with Hoechst 33258 (#H21491; Thermo Fisher) and imaged using a Nikon Eclipse Ti2 to quantify cell number. Normalization was completed using these cell counts.

### 2.6. Western Blotting

Following infection, cells were washed with cold 1× PBS and lysed with RIPA buffer supplemented with phosphatase and protease inhibitors (#88662, #88661; Thermo Fisher). Cells were sonicated and protein concentration of the soluble fraction was determined using a BCA assay (#23235; Thermo Fisher). Proteins were separated on a 4–12% Bis-Tris gel (#NP0322-BOX; Thermo Fisher) and transferred to nitrocellulose using the iBlot system (#IB301002; Thermo Fisher). Membranes were blocked with 5% nonfat dry milk (wt/vol) in 1× TBS with 1% Tween-20. Membranes were then incubated with primary antibodies overnight at 4 °C, washed, and incubated at room temperature for 1 h with anti-mouse (1:2000; #7076; Cell Signaling Technology, Danvers, MA, USA) or anti-rabbit (1:10,000, #7074, Cell Signaling Technology) HRP-conjugated secondary antibodies. Membranes were imaged on an Amersham Imager 600 (General Electric, Boston, MA, USA).

Primary antibodies: anti-GAPDH Rabbit polyclonal antibody (1:1000; #G9545; Sigma Aldrich, St. Louis, MO, USA), anti-HCMV-IE1 mouse polyclonal antibody (1:200; Bill Britt), anti-HCMV-pp28 mouse monoclonal antibody (1:100; #sc-56975; Santa Cruz Biotechnology, Dallas, TX, USA), anti-HCMV-pp52 mouse monoclonal antibody (1:200; #sc-56971; Santa Cruz), anti-HCMV pp65 mouse monoclonal antibody (1:200; #sc-52401; Santa Cruz), anti-MCT1 mouse monoclonal antibody (1:100; #sc-365501; Santa Cruz), anti-MCT4 mouse monoclonal antibody (1:100; #sc-376140; Santa Cruz), anti-LDHA (1:1000; #PA5-27406; Thermo Fisher), anti-LDHB (1:1000; #PA5-96736; Thermo Fisher).

### 2.7. Flow Cytometry

LN-18 cells were mock- or HCMV-infected for the desired timepoints and analyzed by flow cytometry as previously described [31]. Briefly, mitochondrial membrane potential was measured using MitoTracker Deep Red (200 nM; #M22426, Thermo Fisher) or MitoTracker Orange CMTMRos (250 nM; #M7510, Thermo Fisher). Reactive oxygen species production was assessed using CellROX Deep Red (750 nM; #C10491, Thermo Fisher) or CellROX Orange (750 nM; #C10493, Thermo Fisher). Superoxide production was examined using MitoSOX Red (5 μM; #M36008, Thermo Fisher). The representative gating strategy used for these experiments is described in Appendix A. Briefly, singlets were identified and selected for; live cells were gated using Sytox Blue Dead Cell Stain (1:1000; #C10491, Thermo Fisher); GFP expression was used to gate on HCMV infected cells (Towne-GFP); finally, the dye of interest was examined.

For metabolic flow panels, cells were mock- or HCMV-infected for the desired timepoints, incubated with LIVE/DEAD Fixable Dead Cell Stain (1:1000; #L34957; Thermo Fisher) for 30 min at room temperature before fixation and permeabilization using eBioscience Foxp3/Transcription Factor Staining Buffer Set (#00-5523-00; Thermo Fisher) for 20 min at room temperature. Cells were washed with 1× Perm Buffer and incubated with conjugated primary antibodies for 1 h at room temperature. Cells were washed 2× with Perm Buffer and resuspended in a flow buffer for analysis. Data was acquired using a LSRFortessa (BD Biosciences; San Jose, CA, USA) and quantified using FlowJo software (BD Biosciences).

Primary antibodies: anti-MCT1-AF647 (1:100; #sc-365501 AF647; Santa Cruz), anti-MCT4-PE (1:100; #sc-376140 PE; Santa Cruz), anti-Tomm20-AF405 (1:1000; #ab210047; Abcam, Cambridge, UK), anti-H3K27me3-PE (1:1000; #40724; Cell Signaling Technology), anti-GLUT1-AF647 (1:1000; #ab195020; Abcam) anti-VDAC1 (1:1000; #ab14734; Abcam). The VDAC1 antibody was conjugated to PE-Cy5 using the Lightning-Link Kit (#ab1023893; Abcam) according to the manufacturer’s instructions.

### 2.8. Lactate Assay

LN-18 cells were mock- or HCMV-infected for the desired timepoints. The supernatant was collected and clarified by simple centrifugation. Cells were lysed with RIPA buffer as described above. Supernatant and cell lysates were deproteinized with TCA following the manufacturer’s instructions (#ab204708; Abcam). Lactate concentration was determined by comparison with standards using an L-lactate assay kit (#ab65331; Abcam).

### 2.9. Co-Culture Assay

HFF cells were seeded into 6-well plates. LN-18 cells were seeded into the upper portion of a 6-well Transwell with 0.4 μm pores (#3470; Corning) and cultured separately from the HFFs. Once confluent, the LN-18s were infected with HCMV Towne-GFP at an MOI of 3 and incubated for 90 min at 37 °C. Cells were washed twice with 1× PBS, fresh complete culture media was added, then Transwells were placed in the 6-well plates with the HFFs for the desired infection period. At the final timepoint, HFFs were examined for signs of CPE and the cell culture medium was titered as described above to determine the presence of any virus. Cells were isolated and assayed by flow cytometry.

### 2.10. Statistical Analysis

All data are expressed as mean standard error of the mean (SEM) from at least three independent experiments. All data were analyzed using Prism 9 software (GraphPad; San Diego, CA, USA). Comparisons between mock and HCMV groups were assessed using unpaired *t* tests. Ordinary one-way analysis of variance followed by Dunnett’s multiple-comparison test was used to compare samples across time points. For all comparisons, a *p* < 0.05 was considered significant.

## 3. Results

### 3.1. HCMV Infects and Replicates in a Human Glioblastoma Cell Line LN-18

We examined the infection kinetics of HCMV Towne-GFP (IE1-GFP) in the glioblastoma cell line LN-18 to confirm tropism and to characterize its replication cycle. Following infection, RNA, cell lysate, and supernatant were collected daily. Transcriptional expression of HCMV immediate early gene 1 (IE1) was evident at 24 h post infection (HPI) and remained present throughout the time course (Figure 1A). Protein expression of IE1, early gene HCMV-pp65, leaky-late gene HCMV-pp52, and true late gene HCMV-pp28 matched published data defining HCMV gene expression products (Figure 1B) [29]. Consistent with our translational expression data, LN-18 cells exhibited a replication cycle of ~120 HPI as determined by initial cellular lysis and infectious virus release (Figure 1D).

An Agilent xCELLigence platform was used to dynamically monitor longitudinal viral infection of LN-18 cells following infection (Figure 1C). xCELLigence records the Cell Index, a unitless proxy for the cellular impedance of electron flow produced by a confluent cell layer. As the infection progresses, cytopathic effect (CPE) increases as the cells round, decreasing impedance and therefore Cell Index. As evidenced by the Cell Index plot, HCMV infection reduced Cell Index relative to mock-infected cells at 48 HPI. Taken together, the transcriptional and translational detection of HCMV genes, induction of CPE, and release of infectious virus by 120 HPI indicate that HCMV is able to productively infect LN-18 cells.

### 3.2. HCMV Infection Metabolically Reprograms LN-18 Cells

Previous research in HFFs has indicated that HCMV is a potent regulator of host cell metabolism and causes upregulation of aerobic glycolysis [28,29]. To understand the metabolic profile of HCMV infected glioblastoma cells, we employed the Seahorse XFe24 Bioanalyzer and examined ATP production rate and source. The Seahorse Bioanalyzer measures cellular oxygen consumption rate (OCR) and extracellular acidification rate (ECAR) to assess mitochondrial oxidative phosphorylation (OXPHOS) and glycolysis respectively. Briefly, baseline readings were obtained followed by injection of oligomycin to inhibit ATP production by ATP synthase (Complex V) of the electron transport chain (ETC). This allowed for calculation of the mitochondrial ATP production rate. Next, a mixture of rotenone and antimycin A, inhibitors of ETC Complex I and III, was added to cells to abolish any extracellular acidification related to ETC function. ATP production rate exclusively derived from glycolysis was determined by ECAR measurements. Total ATP production rate was then subdivided into mitochondrial (mitoATP) and glycolysis (glycoATP) derived ATP. All measurements were normalized to the cell number.

At 120 HPI, we witnessed a significant increase in total ATP production rate in HCMV infected cells (Figure 2A). Significant increases in both glycoATP (Figure 2B) and mitoATP (Figure 2C) production rates were also observed.

Interestingly, there was a significant increase in the percentage of glycoATP and a corresponding decrease in the percentage of mitoATP at 72 HPI (Appendix A). Overall, the mitoATP:glycoATP ratio, termed XF ATP Rate Index, favored ATP production by OXPHOS in mock- and HCMV-infected cells (Appendix A). We conclude that HCMV infection induces significant metabolic reprogramming to increase ATP production largely irrespective of the pathway (aerobic glycolysis versus OXPHOS). Presumably, this reprogramming is necessary to meet the increased metabolic and bioenergetic demands of viral replication.

### 3.3. HCMV Infection Increases Oxidative Phosphorylation of Host Cells

To further examine the increased mitoATP production rate, we evaluated mitochondrial function using the Seahorse Bioanalyzer. Briefly, basal respiration rate was assessed via baseline OCR measurement of mock- or HCMV-infected cells (Figure 3A). The cells were treated with oligomycin to inhibit ATP synthase which indicates the respiration linked to ATP production. Next carbonyl cyanide-4 phenylhydrazone (FCCP) was used to disrupt the proton gradient and mitochondrial membrane potential. With maximal ETC complex I-IV function, OCR was used to determine the maximal respiration rate of the mitochondria and the spare respiratory capacity. Finally, rotenone and antimycin A, inhibitors of ETC complexes I and III, were added to determine oxygen consumption unrelated to mitochondria function.

HCMV infection produced an increase in mitochondrial respiration in all treatment conditions (Figure 3B). We saw increases in basal respiration relative to mock-infected cells and maximal respiration capacity (Figure 3C,D). This increase in maximal respiration also indicates a significant increase in spare respiratory capacity of HCMV infected cells (Figure 3E; Appendix A). These results suggest that HCMV infection has increased host cell mitochondrial function permitting increased energetic production as observed in Figure 2.

In agreement with the previous ATP Rate Assay, HCMV-infected cells exhibit significantly increased ATP production (Appendix A). Additionally, host cells had elevated basal respiration accounting for a greater percentage of their maximal respiratory capacity. This suggests that host cells are increasing mitochondrial respiration to produce the biosynthetic materials and bioenergy necessary for viral replication.

Proton leak, a measure of mitochondrial damage or alternatively a means of regulating mitochondrial membrane potential, was significantly increased in HCMV infected cells relative to controls (Figure 3E). Finally, there were no significant changes in non-mitochondrial oxygen consumption or coupling efficiency with HCMV infection at either timepoint (Appendix A).

### 3.4. HCMV Infection Increases Mitochondrial Membrane Potential and Production of Reactive Oxygen Species

Upregulation of OXPHOS can result in increased mitochondrial membrane potential as the mitochondria attempt to meet energetic demands. To investigate HCMV-mediated alterations in membrane potential we labeled cells with MitoTracker Orange CMTMRos, a reduced rosamine-based dye which, when oxidized by oxygen in respiring cells, becomes fluorescent. At all timepoints following infection, HCMV-infected cells exhibited increased membrane potential relative to their mock-infected counterparts with a peak at 96 HPI (Figure 4A). To confirm these findings, we used another membrane-potential-dependent dye, MitoTracker Deep Red. MitoTracker Deep Red is an inherently fluorescent carbocyanine-based dye that has a high affinity for mitochondria. Following infection with HCMV, cells stained with MitoTracker Deep Red exhibited increased mitochondrial membrane potential relative to their mock-infected controls which became statistically significant at 96 and 120 HPI. Despite the differences in mechanism of action, both dyes produced similar results indicating increased membrane potential in HCMV-infected cells (Appendix A).

Upregulation of OXPHOS can result in the production of reactive oxygen species (ROS) that exceeds the neutralization ability of the antioxidant systems including nicotinamide adenine dinucleotide (NAD/NADH). At low levels, ROS acts in a variety of intra- and inter-cellular signaling cascades. However, excessive ROS can cause significant cellular oxidative stress resulting in DNA, protein, and lipid damage and potentially cell death [39]. We examined ROS production using CellROX Deep Red, a dye that becomes fluorescent following oxidation by ROS. We detected increases in ROS species which peaked at 96 HPI (Figure 4B). These findings were confirmed using another oxidation dependent dye, CellROX Orange (Appendix A).

During respiration, superoxide can be produced from specific mitochondrial sites. We employed MitoSOX Red, a dye that becomes fluorescent following oxidation by superoxide but is resistant to oxidation by other ROS. Similar to the CellROX results, we witnessed significant increases in superoxide production at 48 HPI and at all subsequent timepoints (Figure 4C). These increases in superoxide were not observed in the mock-infected cells. These findings suggest that, following infection, the ETC is running at an elevated rate and producing superoxide at a rate that surpasses the neutralization ability of antioxidant pathways.

### 3.5. HCMV Infection Increases Glycolytic Rate and Capacity

The ATP Rate Assay indicated increased glycoATP and the percentage of ATP derived from glycolysis. To further investigate this, we again employed the Seahorse Bioanalyzer. Briefly, ECAR measurements were taken for both mock- and HCMV-infected cells prior to the introduction of glucose (Figure 5A). Measurements obtained in the presence of glucose established the baseline rate of glycolysis. Next, oligomycin was added to inhibit ATP synthase and eliminate all mitochondrial ATP production, forcing the cells to depend on glycolysis for energy production. This provided the maximal glycolytic capacity as well as the glycolytic reserve available. Finally, 2-deoxy-d-glucose, a glucose analog and inhibitor of hexokinase-2, was added to inhibit glycolysis and identify ECAR unrelated to glycolysis.

HCMV infection resulted in increased glycolysis in all treatment conditions (Figure 5B). HCMV-infected cells exhibited increased basal and maximal glycolytic rate relative to mock-infected cells (Figure 5C,D). These results suggest that HCMV has started to alter the glycolytic behavior of the cells and is consistent with previous studies of HCMV infection indicating increases in glycolysis due to upregulation of GLUT4 [29].

There was a trend for HCMV-infected cells to use more of their maximal glycolytic capacity relative to mock-infected cells as evidenced by decreased percentage of glycolytic reserve (Figure 5E). Finally, there were no significant differences in non-glycolytic acidification or glycolytic reserve between mock- and HCMV-infected cells, although both trended toward an increase with HCMV infection. Taken together, these findings suggest that HCMV reprograms host cells to respond to increased energetic needs by increasing basal- and the potential for maximal-glycolysis relative to mock-infected cells. Thus, we decided to examine the effects of increased aerobic glycolysis on lactate production.

### 3.6. Elevated Aerobic Glycolysis following HCMV Infection Alters Lactate Production and Flux

Our Seahorse Bioanalyzer data indicated significant increases of glycolysis following infection with HCMV. One product of aerobic glycolysis is lactate; thus, we employed an L-lactate assay kit to examine both the intracellular (Figure 6A) and extracellular (Figure 6B) concentrations of lactate. Intracellular lactate was increased in HCMV-infected cells from 48 HPI through 168 HPI relative to mock-infected cells. This correlates well with the enhanced glycolysis and glycoATP production previously discussed at 72 and 120 HPI. We also observed significantly increased extracellular lactate at 144 HPI and 168 HPI in the supernatant of HCMV-infected cells relative to mock-infected cells. These findings suggest that lactate production and homeostasis were significantly altered following HCMV infection. Increased lactate concentrations also agree with previously published work in MRC5 cells following HCMV infection [40].

Lactate, in addition to being an end product of glycolysis, is a potent signaling molecule that can function in both a paracrine and an autocrine manner. As a result, lactate homeostasis is an important mechanism within cells. Initially, we investigated the transcriptional changes of key lactate genes following infection with HCMV. Surprisingly, we observed a downregulation of gene expression for several lactate enzymes and transporters relative to mock-infected controls (Appendix A). These results were reiterated by RNA-seq data from mock- or HCMV-infected G44 GBM cells at 72 HPI. However, when we examined the protein expression following HCMV infection we found surprising changes (Figure 6C). Lactate dehydrogenase (LDH)-A, responsible for the conversion of pyruvate to lactate, expression remained stable throughout infection. However, LDH-B, responsible for the conversion of lactate to pyruvate, decreased at later timepoints. Two monocarboxylate transporters, MCT1 and MCT4, responsible for lactate-influx and -efflux, respectively, were examined. HCMV-infected cells demonstrated decreases in MCT1, but MCT4 increased relative to mock-infected cells. The protein expression changes cumulatively suggest a lactate management phenotype that favors the production of lactate from pyruvate. It also suggests that elevated intracellular lactate concentration increases MCT4 expression which provides a mechanism for export and may allow for improved lactate homeostasis.

Co-culture of HCMV-infected LN-18 with HFFs alters the expression of key metabolic proteins and epigenetic markers in both cell lines. In an effort to examine the effects of HCMV-infected LN-18 cells on neighboring non-cancerous cells, we established physically separated co-cultures with HFFs using 0.4 μm pore Transwells (Figure 6D). LN-18s were seeded in Transwells and infected with HCMV for two hours followed by washing and refeeding. At this point the LN-18 Transwells were placed in a 6-well plate containing confluent HFFs. This allowed for paracrine communication between cell lines while keeping them physically separated. Flow cytometry demonstrated HCMV-GFP positivity in the LN-18s and an absence of HCMV-GFP positivity at the longest co-cultured timepoint, 120 HPI, in HFFs (Figure 7A). Additionally, HFFs were confirmed negative for CPE and their supernatant was titered and found free of the virus at the longest co-culture timepoint, 120 HPI (Figure 7B).

Following co-culture, infected LN-18 cells exhibited a trend towards decreased MCT1 expression but failed to reach significance (Figure 6E). They also demonstrated a significant increase in MCT4 expression (Figure 6G). In contrast, the HFFs had increased MCT1 expression at the earliest timepoint which decreased significantly at 96 and 120 HPI (Figure 6F). HFF expression of MCT4 remained largely unchanged (Figure 6H). These findings suggest that the HCMV-mediated increase in glycolysis in LN-18 cells causes a buildup of lactate intracellularly. This lactate needs to be exported presumably to avoid cytotoxicity, thus the expression of lactate efflux transporter, MCT4, increases to maintain intracellular homeostasis. HFFs, being presented with increased extracellular lactate concentrations, downregulate their lactate importer, MCT1, to maintain their own homeostasis.

Based on the MCT expression changes following co-culture, we used the same experimental method to examine several metabolic and transcriptional targets. Glucose transporter 1 (Glut1), the primary glucose importer of cells, was found to be upregulated at all timepoints in both LN-18 (Figure 7C) and HFF (Figure 7G) cells. Mitochondrial import receptor subunit TOM20 homolog (Tomm20), a key mitochondrial import receptor, was found to be increased in HCMV-infected LN-18 cells (Figure 7D). In HFFs there was a similar increase at early timepoints followed by decreased expression at the latest timepoint (Figure 7H). the voltage gated mitochondrial membrane ion channel (VDAC1) is responsible for the transport of ions, nucleotides, and metabolites through the mitochondrial membrane and was upregulated in LN-18 cells at the last timepoint (Figure 7E). However, in HFFs, VDAC1 was upregulated at the earliest timepoint and subsequently downregulated at the final two timepoints (Figure 7I). Taken together, these expression changes provide support for the hypothesis that HMCV infected cells can alter the metabolic function of neighboring cells in a paracrine manner akin to the Reverse Warburg Effect described in the cancer field [34].

Evidence of paracrine signaling-mediated effects on neighboring uninfected cells suggests that HCMV infection may also be able to alter the transcriptional landscape of both host and neighboring cells. We examined tri-methylation of histone H3 lysine 27 (H3K27me3) as an indicator of epigenetic regulation following infection with HCMV. Hypermethylation is a potent repressor of gene expression. Histone 3 lysine 27 transcriptional repression has been linked to regulation of cell differentiation and proliferation [41]. Numerous studies have linked alterations in H3K27me3 to the development and progression of cancer [42,43,44]. In the HCMV-infected LN-18 cells, there was a significant increase in methylation at the first and two final timepoints (Figure 7F). Additionally, there was significant upregulation of H3K27 tri-methylation of the HFF cells at the first timepoint. These findings suggest that HCMV infection produces epigenetic changes in the host cells that may favor tumor progression. Additionally, it indicates that neighboring cells are also susceptible to epigenetic modifications as a result of an unidentified paracrine messenger.

## 4. Discussion

HCMV has been implicated as a potential oncomodulatory virus in GBM and some other tumor types [9,38,45]. Defining a role for HCMV in human cancers has been hindered by numerous factors. In GBM specifically, conflicting results on the presence of HCMV in primary GBM samples have created justified questioning of significance (reviewed in [45]). A lack of reproducibility between laboratories employing similar techniques suggests that current experimental approaches are inadequate. Identifying a relationship is further complicated by the long period between initial HCMV infection and the initiation or manifestation of cancer. Yet, under in vitro conditions, HCMV can alter or influence all the hallmarks of cancer. In vivo murine GBM studies displayed reactivation of MCMV in the tumor and promotion of tumor growth [38]. Preliminary clinical studies using herpesvirus antivirals in addition to the standard of care demonstrated increased life expectancy in patients with GBM [19,46]. These observations collectively suggest a role for HCMV in GBM. Our work and others have shown that HCMV infection rewires host metabolic pathways to meet biosynthetic and bioenergetic requirements of viral replication [29,31,40]. Dysregulated cellular energetics is an emerging hallmark of cancer [33]. Thus, viral manipulation of host metabolism represents a potential mechanism underlying the influence of HCMV on tumor progression in GBM.

Our work presented here shows that HCMV alters the metabolic phenotype of an infected GBM cell line. This does not require a transformation of the cells. We observed upregulation of glycolysis and mitochondrial respiration following infection with HCMV. These results are similar to the metabolic reprogramming previously demonstrated in HFF cells [29,31,32]. They also resulted in the production of metabolic byproducts, such as ROS and secreted metabolites such as lactate, which have been shown to play roles in cancer progression. Further, infected cells exhibited alterations in epigenetic markers associated with cancer initiation. Finally, we demonstrated that HCMV infected cells were able to manipulate key metabolic pathways of neighboring cells in a paracrine manner independent of viral transmission. These findings suggest a role for HCMV-mediated metabolic reprogramming in the manipulation of the tumor microenvironment (TME) potentially leading to cancer progression.

Upregulation of oxidative phosphorylation (Figure 3) during HCMV infection of GBM cells suggests that HCMV may be metabolically synergistic, driving metabolic rates above what is possible following oncogenesis alone. GBM has been shown to display high OXPHOS levels in vivo [47]. Glioblastoma stem-like cells (GSCs) and surgically resected tumors from GBM patients use oxidative and non-oxidative pathways to generate energy and biomass [47,48]. It is possible that HCMV infection enhances metabolic potential allowing for the maintenance of high proliferation rates seen in GBM. Increases in spare respiratory capacity (Figure 3, Appendix A) observed following HCMV infection suggest that there are significant bioenergetic reserves soon after HCMV infection. GSCs are reliant on OXPHOS for proliferation and have been proposed as a potential therapeutic target [49]. Increases in both OXPHOS and bioenergetic reserves highlight the potential for HCMV infection to allow for increased growth rate, metabolic flexibility, and potential for drug resistance in GBM cells.

Alternatively, as a consequence of upregulated OXPHOS, we observed increased production of ROS (Figure 4), specifically superoxide, following infection of GBM cells. High levels of oxidative stress can damage proteins, lipids, and DNA further contributing to conditions conducive for cancer initiation or progression (reviewed in [50,51]). Specifically, ROS mediated oxidation of phosphatase and tensin homolog (PTEN) causes a loss of function and activates the phosphoinositide 3-kinase (PI3K), protein kinase B (Atk), mammalian target of rapamycin (mTOR) signaling cascade, leading to protein synthesis and proliferation [52]. Sustained proliferative signaling is one of the hallmarks of cancer and genomic instability and mutation is described as an enabling characteristic of cancer [33]. An examination of HCMV-associated alterations to PI3K/Akt/mTOR signaling following infection could yield intriguing insights in relation to cancer. Additionally, this mechanism could point towards the use of an ETC inhibitor such as metformin to control the production of ROS and subsequent activation of the PI3K/Akt/mTOR signaling cascade. We recently reported that ETC inhibitors, specifically metformin, can negatively impact HCMV replication [53].

Many of the metabolic changes closely resemble metabolic alterations reported in cancer such as the upregulation of aerobic glycolysis (Figure 5 and Figure 8). GBM is dependent on a functioning glycolytic cycle with targeted knockdown of glycolytic enzymes producing increased longevity in a murine model of GBM [54]. The HCMV protein US28 has been demonstrated to increase hypoxia inducible factor-1 (HIF-1)-dependent proliferation of GBM cell line U251 through upregulation of glycolysis [55]. Therefore, the increased glycolytic capacity and rate that we see following infection with HCMV may support enhanced tumor growth rates. In the context of GBM, increased Glut1 expression may allow for enhanced glucose uptake which may help compensate for poor vascularization and nutrient dispersal in the tumor microenvironment. This may allow GBM cells to maintain their characteristic growth and proliferation rates despite environmental stress.

Following glycolytic conversion of glucose to pyruvate, enzymes can convert pyruvate to lactate. Previous studies have shown that HCMV infection increased the concentration of L-lactate in the supernatant of cultured U251 GBM cells [55]. We observed changes to lactate levels (Figure 6) in both intracellular and extracellular compartments relative to mock-infected cells. Lactate secretion into, and acidification of, the TME has been shown to promote GSC phenotypes, induce angiogenesis, and inhibit immune responses [56]. Notably, significant increases to lactate exporter MCT4 expression were seen (Figure 6 and Figure 8, Appendix A) and may account for the increased extracellular concentrations of lactate MCT4 expression is elevated in GBM, has been positively correlated with the pathological grades of glioma, and was found to promote GBM migration [57]. Therefore, the HCMV-mediated increases in MCT4 levels we witnessed may prove beneficial to tumor progression and metastasis.

The reverse Warburg Effect has been proposed as a mechanism that exploits the metabolic potential of cells in the TME for the benefit of cancer stem cells [36]. In addition to its role acidifying the TME, extracellular lactate can also act as a fuel source for oxidative cancer cells and cells with stem-like phenotypes [58,59]. Reciprocal metabolic associated interactions involving catabolism of glucose, lactate, fatty acids and other metabolites impact cancer aggressiveness. This suggests that HCMV infection of GBM cells could alter the surrounding TME via the secretion of metabolites or proteins in a mechanism similar to that employed by tumors to promote growth and invasion. Our co-culture experiments (Figure 7) show that this is possible and that the infected secretome can alter metabolism-associated and epigenetic targets. Notably, hypermethylation of H3K27 has been linked to the development and progression of cancer [41]. Epigenetic modifications have the potential to alter the transcription of large groups of genes and can produce potent phenotypic changes in cells. HCMV-mediated epigenetic alterations in both infected and neighboring cells suggest the potential for a small number of cells to have exponential effects within the TME. It is unknown, at this time, if these epigenetic alterations promote a tumor-like phenotype, can prime surrounding cells for infection, or rewire metabolism in infected and uninfected cells. It will be interesting to explore what mechanistic effect the secretome is having and how this affects the long-term state of the tumor cells. Targeted experiments identifying secreted metabolites, signaling proteins, or extracellular vesicles are currently underway in the laboratory. GSCs have been shown to communicate to the tumor microenvironment through extracellular vesicles (reviewed in [60]). It will be intriguing to understand how HCMV infected cells and GSCs communicate and if this communication drives synergistic effects that promote tumor growth and metastasis.

There are several limitations to our study. We used a GBM cell line that is immortalized and grown as a monolayer using high glucose media in supraphysiological oxygen. As we compared changes with and without HCMV infection, with all other variables controlled for, we believe that this caveat is negligible. Regardless, ongoing work in the laboratory is employing primary GSCs grown under serum-free conditions. Furthermore, we did not explore all major metabolic pathways known to be affected by HCMV infection. HCMV infection has been shown to block beta-oxidation of fatty acids in order to promote the lipogenesis necessary to produce the viral envelope [61]. Human glioma cells have been reported to be dependent on fatty acid oxidation under tissue culture conditions [62]. We also have not explored the role of glutamine as cultured astrocytes have been shown to support the growth of glutamine deprived GBM cells [63]. Our model ignores any contributions of oxygen tension which are prevalent in the context of the TME. There is ample literature describing the metabolic interplay between hypoxic and normoxic tumor microenvironments and how this dictates catabolic versus anabolic behavior. Hypoxic conditions also increase the expression of MCT4 [64]. Finally, we do not account for the relationship between increasing HCMV seroprevalence and GBM incidence with advancing chronologic age. Integrating this correlation into our experimental approach would be intriguing.

Our results provide several potential mechanisms by which HCMV could contribute to GBM and possibly other cancers. It also offers insight into the difficulty of detecting HCMV in GBM samples. First, HCMV would only need to infect a small number of cells to exert a significant effect on the tumor microenvironment. Second, our data suggest that HCMV infected cells can employ paracrine signaling to modify neighboring cells independent of infection, thus not requiring direct infection of GBM cells to produce significant changes. The ability of HCMV infected cells to transiently alter epigenetic signatures in neighboring cells hints at the possibility of a role in cancer initiation. At this point, it is unknown if this mechanism has the potential to be cancer-initiating or merely promote the progression of established GBMs. The metabolic reprogramming we demonstrated following infection compliments a study by Liu et al. that showed HCMV infected primary GBM cells exhibit a stem cell like phenotype which enhances their ability to resist chemotherapy [65]. Our work establishes a framework for future studies defining how HCMV can contribute to GBM through an oncomodulatory mechanism. The manipulation of host metabolic pathways may prove to be a common mechanism linking not only HCMV, but other pathogens, to numerous cancers. These insights could be used to develop therapies that incorporate metabolic targets in combination with the current standard of care.

## Figures and Tables

**Figure 1 viruses-14-00103-f001:**
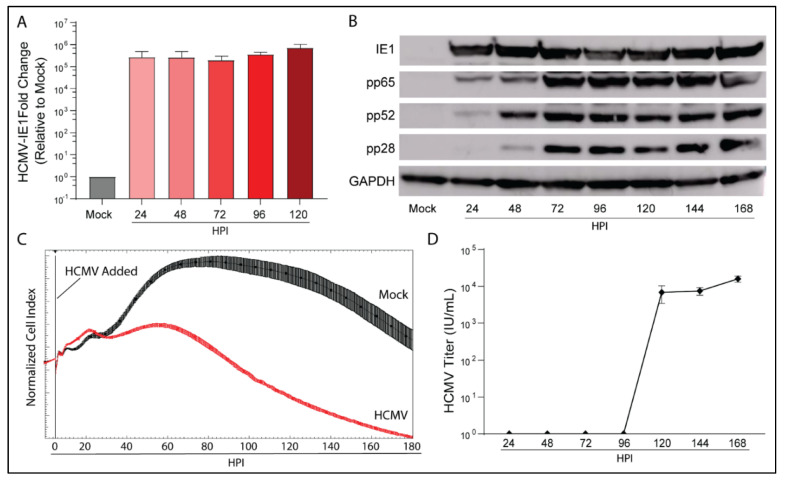
HCMV can productively infect the glioblastoma cell line LN-18. LN-18 cells were infected with HCMV Towne strain and (**A**) transcriptional and (**B**) translational profiles of key HCMV proteins were examined by qPCR and WB. Mock infected cells served as controls. (**C**) Dynamic monitoring of LN-18 Cell Index (impedance) during HCMV infection using xCELLigence. (**D**) Viral titers of cell culture supernatant were quantified using an immunofluorescent based assay. Graphs represent pooled data from three independent experiments. Mean ± the SEM of at least 3 triplicates.

**Figure 2 viruses-14-00103-f002:**
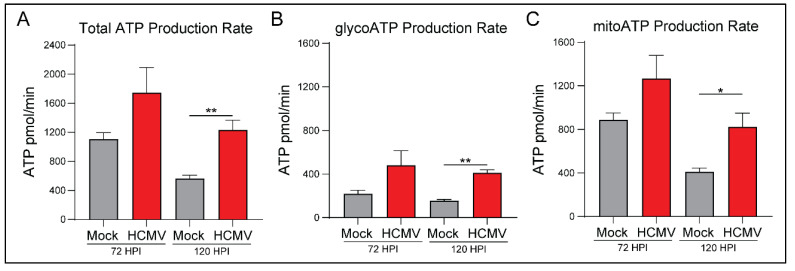
HCMV infection of LN-18 cells increases host ATP production using glycolysis and oxidative phosphorylation metabolic pathways. (**A**) Total ATP, (**B**) glycoATP, and (**C**) mitoATP production rates were quantified at 72 and 120 h post infection using the Seahorse XFe24 Real-Time ATP Rate kit. All samples were normalized to cell number. Graphs represent pooled data from three independent experiments. Mean ± the SEM of at least 3 triplicates. *, *p* < 0.05; **, *p* < 0.01.

**Figure 3 viruses-14-00103-f003:**
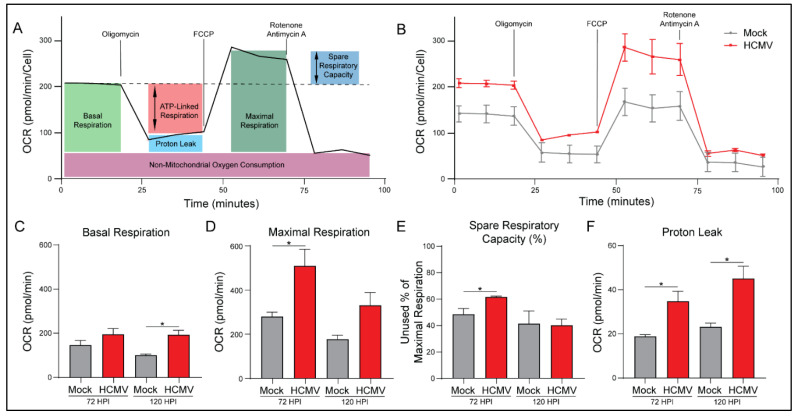
HCMV infection increases oxidative phosphorylation in LN-18 cells. (**A**) A cartoon illustrating how mitochondrial function was measured at 72 and 120 h post infection (HPI) using the Seahorse XFe24 MitoStress Kit. (**B**) Representative data output of LN-18 cells mock- or HCMV-infected at 72 HPI. (**C**) Basal Respiration, (**D**) Maximal Respiration, (**E**) Spare Respiratory Capacity, and (**F**) Proton Leak were derived from these measurements. All samples were normalized to cell number. Graphs represent pooled data from three independent experiments. Mean ± the SEM of at least 3 triplicates. *, *p* < 0.05.

**Figure 4 viruses-14-00103-f004:**
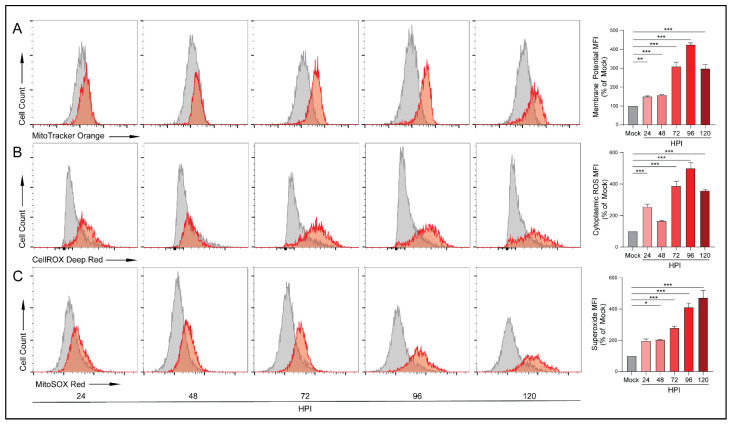
HCMV infection increases electron transport chain function. Flow cytometry was used to quantify functional aspects of the electron transport chain. (**A**) Mitochondrial membrane potential was analyzed in mock- or HCMV-infected LN-18 cells using the reduced rosamine-dye, MitoTracker Orange which is dependent on mitochondrial oxidation to become fluorescent (**B**) cytoplasmic ROS production was detected by CellROX Deep Red, and (**C**) mitochondrial superoxide levels were measured using MitoSOX Red. All samples were gated on singlets and live cells only. Graphs represent pooled data from three independent experiments. Mean ± the SEM of at least 3 triplicates. *, *p* < 0.05; **, *p* < 0.01; ***, *p* < 0.001.

**Figure 5 viruses-14-00103-f005:**
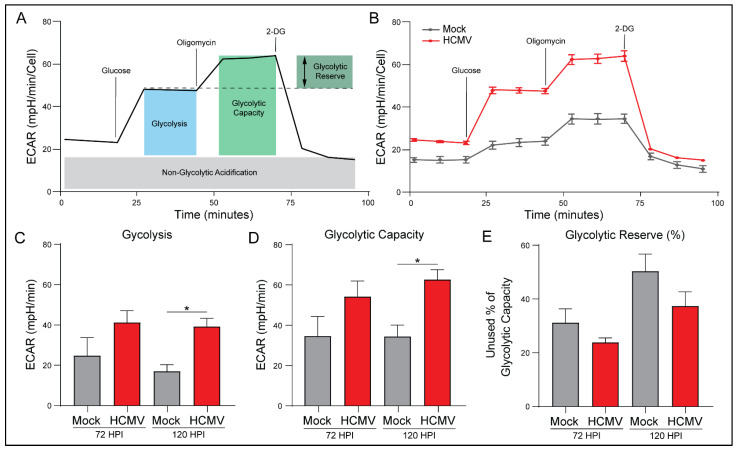
HCMV infection increases glycolysis and glycolytic capacity. (**A**) Measurements of the extracellular acidification rate (ECAR) determined using a Seahorse XFe24 Glycolysis Stress Kit are illustrated. (**B**) Representative data output of LN-18 cells mock- or HCMV-infected. (**C**) Glycolysis, (**D**) Glycolytic Capacity, and (**E**) Glycolytic Reserve were derived from these measurements. All samples were normalized to cell number. Graphs represent pooled data from three independent experiments. Mean ± the SEM of at least 3 triplicates. *, *p* < 0.05.

**Figure 6 viruses-14-00103-f006:**
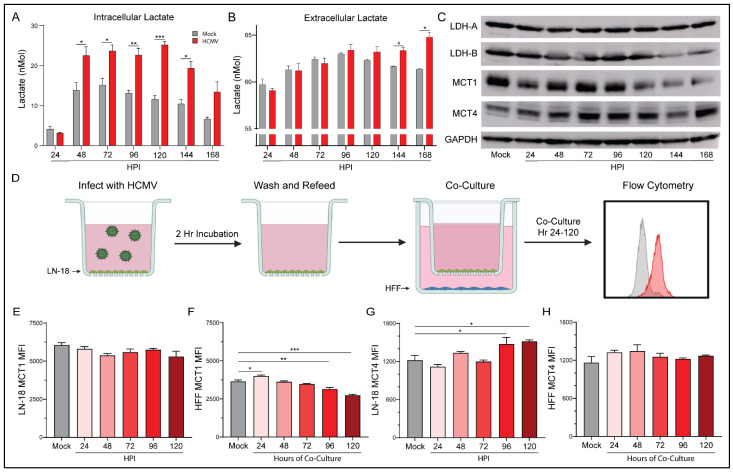
HCMV-associated alterations to lactate dynamics. Lactate concentration was determined in the (**A**) extracellular and the (**B**) intracellular compartment of mock- or HCMV-infected LN-18 cells using a colorimetric L-lactate assay kit with a standard. (**C**) Protein expression of prominent lactate regulators LDH-A, LDH-B, MCT1, and MCT4 were examined following infection using Western blot assays. (**D**) Schematic of co-culture experiment in which HCMV-infected LN-18 cells were co-cultured, but physically separated from, HFF cells using Transwells with 0.4 μm pores in 6-well culture dishes. At indicated points post infection, cells were collected separately and processed by flow cytometry for expression of MCT1 (**E**,**F**) and MCT4 (**G**,**H**) in infected LN-18 and non-infected HFF cells respectively. Graphs represent pooled data from three independent experiments. Mean ± the SEM of at least 3 triplicates. *, *p* < 0.05; **, *p* < 0.01; ***, *p* < 0.001. Created with BioRender.com.

**Figure 7 viruses-14-00103-f007:**
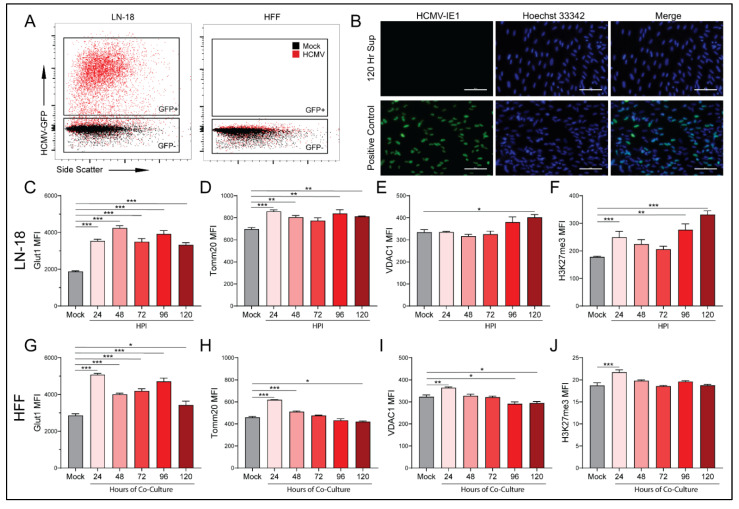
HCMV infected cells alter metabolic pathways of uninfected cells in a paracrine manner. Mock- and HCMV-infected LN-18 cells were co-cultured with HFFs as described in Figure 6D and then the separate populations were subjected to flow cytometry. Infection of LN-18 cells and absence of infection of HFFs was established following 120 Hr co-culture by (**A**) HCMV-GFP expression. A titer was performed on fresh HFFs using supernatant from the 120 Hr co-culture HFF compartment, to ensure an absence of viral infection, and cells which had been infected with HCMV as a positive control for the immunocytochemistry staining. (**B**) HCMV-IE1 staining was performed at 20× magnification. HCMV-IE1 is shown in green and Hoechst 33342 stained nuclei are shown in blue. Scale bar: 100 μM. Flow cytometry was used to quantify changes in glycolytic marker (**C**,**G**) Glut1, oxidative phosphorylation markers (**D**,**H**) Tomm20 and (**E**,**I**) VDAC1, and an epigenetic marker (**F**,**J**) H3K27me3 in infected LN-18 and non-infected HFF cells respectively. Graphs represent pooled data from three independent experiments. Mean ± the SEM of at least 3 triplicates. *, *p* < 0.05; **, *p* < 0.01; ***, *p* < 0.001.

**Figure 8 viruses-14-00103-f008:**
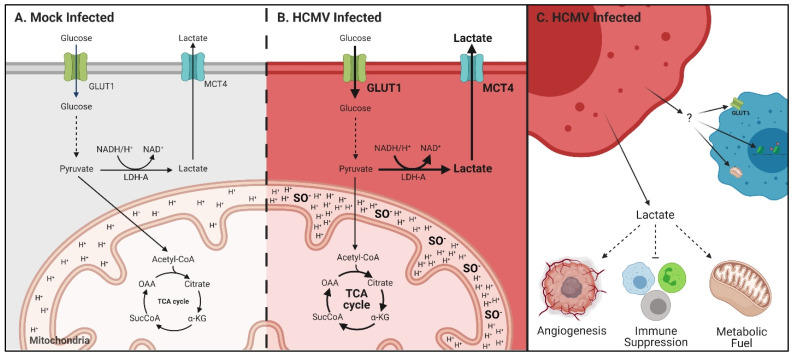
Metabolic reprogramming and paracrine effects of HCMV infection in glioblastoma. Schematic of glycolysis and respiration in (**A**) mock- and (**B**) HCMV-infected cells highlighting upregulation of GLUT1 and MCT4 expression; increased lactate production and export; enhanced respiration and resultant increases in mitochondrial membrane potential and superoxide production. (**C**) HCMV infected cells export lactate which has been reported to increase angiogenesis, suppress the immune response, and can be used as a metabolic fuel by neighboring cells. The secretome from HCMV-infected cells can influence transporter expression, mitochondrial function and induce epigenetic changes in surrounding stromal cells. Created with BioRender.com.

## Data Availability

The data presented in this study are available on request from the corresponding author. The data are not publicly available.

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
