# Peer review of "Metabolic Reprogramming of Glioblastoma Cells during HCMV Infection Induces Secretome-Mediated Paracrine Effects in the Microenvironment"

_viruses, 2022, doi:10.3390/v14010103_

Round 1

Reviewer 1 Report

Harrison and colleagues report analyses of the metabolic effect of HCMV in a glioblastoma cell line model.  They find that infection stimulates metabolism, including OXPHOS and glycolysis, and intracellular and extracellular lactate levels, similar to what has been reported in earlier studies in human fibroblasts (HF). More interestingly, they also find paracrine effects of HCMV infection of GBM cells on uninfected HF cultured separately but in the same transwells.  Their data support that hypothesis that a small number of infected cells in a tumor might have onco-modulatory effects on nearby cells.

Major Comments:

The possible role of HCMV in GBM is controversial, as the authors point out in the text.  However, the wording in the abstract makes it sound like HCMV has a proven link to GBM and I think the wording here should be revised to correct that impression.

The authors cite several older papers that examined metabolic effects of HCMV in other cells such as HF, but they should clarify which of the changes they report in the GBM cells are the same or different from HF.  It seems to me the that changes are very similar to what has been reported for HF. For example, increases in glycolysis and lactate excretion were reported years ago, making these findings a bit less interesting.

There seems to be a bit of apparent redundancy in the presentation of the metabolic data. The text could be more succinct in discussing the inconsequential minor differences between two time points in several experiments. The authors could explain more clearly what is different between panels in Figs. 2 and 3 vs. Supp. Fig. 2? For example, Fig 2B and Supp Fig. 2B seems very similar?  Also, are Fig 3E and Supp Fig. 2C different plots (normalized vs not) of the same data?

Minor comments:

On pages 6 (line 256) and 7 (line 266), the references to panels C and D of Figure 1 are reversed compared to what is on the figure.

The legend to Fig. 2 says that panel A shows a cartoon, but a cartoon more like the one in Fig. 3A would be better than what is shown.

Fig. 3B.  What time point post infection is shown here?

Line 486.  “as a positive control” – not “and a positive control”?

The legend for Fig 7B should clarify which cell type supernatant (HF vs LN-18)
 was used.

Line 562. What does metabolically synergistic mean?  Synergistic with what?

Line 652. The role of high oxygen may not be negligible.

Reviewer 2 Report

The authors studied the metabolic programming of glioblastoma during human cytomegalovirus (HCMV) infection. This is an interesting study to determine the potential oncomodulatory effect of HCMV. They demonstrated the metabolic changes following viral infection using a GBM cell line (LN-18) and a laboratory strain of HCMV (Towne) showing increased mitochondrial respiration, glycolytic rates, and lactate production following viral infection. They further show the impact on the tumor microenvironment using a GBM-fibroblast coculture model.

Specific Comments:

  • qPCR data in Fig 1A was shown relative to mock, where mock is not expected to produce any signal. Transcriptional expression versus mock can be used (for instance) for host genes, where the infected condition can be compared with mock. It would have been more appropriate to quantify the virus by including a standard curve in the assay.
  • The use of a clinical isolate could have strengthened the study.
  • Fig 1C (line 256) and Fig 1C (line 266) refer to the wrong panel.
  • The inclusion of schematic in Fig 2a, 3a, 5a, and 6d has helped illustrate the experimental design or measurements. However, there is no need to show mock and HCMV in the cartoon in Fig 2a if it is not the actual data. Only one bar showing glycoATP, mitoATP, and TotalATP may serve the purpose.
  • Line 94: Do authors mean the Reverse Warburg effect instead of Warburg effect while citing Pavlides et al. 2019? The reverse Warburg effect hypothesis would mean increased aerobic glycolysis and lactate production in fibroblasts, while the authors have not shown this. Instead, this was shown in GBM cells.
  • The reverse Warburg hypothesis has its place in discussion but not in the title. The title can be more descriptive, instead of “Reverse Warburg Effect-Like Changes”.
  • The discussion section is relatively long and may be revised to be more focused.
  • Line 542: MCMV instead of MHCMV
  • Has the RNA-seq data been deposited in a publicly accessible repository (GEO/SRA)?
